# Contemporary Art Authentication with Large-Scale Classification

Todd Dobbs [1] , Abdullah-Al-Raihan Nayeem [1] , Isaac Cho [2] and Zbigniew Ras [1,3,*]

1 Department of Computer Science, University of North Carolina at Charlotte, Charlotte, NC 28223, USA; btdobbs@uncc.edu (T.D.); anayeem@uncc.edu (A.-A.-R.N.)
2 Department of Computer Science, Utah State University, Logan, UT 84322, USA; isaac.cho@usu.edu
3 Department of Computer Science, Polish-Japanese Academy of Information Technology, 02-008 Warszawa, Poland
* Correspondence: ras@uncc.edu

**Abstract:** Art authentication is the process of identifying the artist who created a piece of artwork and is manifested through events of provenance, such as art gallery exhibitions and financial transactions. Art authentication has visual influence via the uniqueness of the artist's style in contrast to the style of another artist. The significance of this contrast is proportional to the number of artists involved and the degree of uniqueness of an artist's collection. This visual uniqueness of style can be captured in a mathematical model produced by a machine learning (ML) algorithm on painting images. Art authentication is not always possible as provenance can be obscured or lost through anonymity, forgery, gifting, or theft of artwork. This paper presents an image-only art authentication attribute marker of contemporary art paintings for a very large number of artists. The experiments in this paper demonstrate that it is possible to use ML-generated models to authenticate contemporary art from 2368 to 100 artists with an accuracy of 48.97% to 91.23%, respectively. This is the largest effort for image-only art authentication to date, with respect to the number of artists involved and the accuracy of authentication.

**Keywords:** art authentication; deep learning; digital image processing; machine learning; residual neural network



## 1. Introduction

Proof of an artwork's authenticity is imperative when buying or selling a piece of art. The authenticity of an artist and their work is normally accomplished through a process of documenting a certificate of authenticity, past ownership, artist signature, and other physical attributes, such as dimension, medium, and title. This process is known as the artwork's provenance. As early as 2004, digital techniques for art authentication were conceived to augment physical authentication techniques by analyzing consistencies and inconsistencies in the first- and higher-order wavelet statistics collected from drawings or paintings. Results confirmed the proper authentication of 13 drawings and paintings, either by Pieter Bruegel the Elder or Perugino [1]. In 2017, hyperspectral imaging combined with advanced signal processing techniques correctly identified two Beltracchi forgeries by correctly classifying 78% of pigments in the forged paintings [2].

In the past five years, art authentication received increased attention due to artificial intelligence, digital image processing, forensic techniques, and legal cases. From an artificial intelligence perspective, supervised deep learning algorithms, when applied to images of paintings, have attained an accuracy of 67.78% in authenticating art for 90 artists using the WikiArt dataset [3], and an accuracy of 32.40% in authenticating art for 1199 artists using the Rijksmuseum dataset [4]. On the digital image processing front, an accuracy of 91.7% was achieved in authenticating art for two artists using the principal component analysis (PCA) and a custom van Gogh and Raphael dataset. These results involve fewer artists with the advantage of reduced resource costs [5]. An accuracy of 88% was achieved for

authenticating art on an undisclosed number of artists using a decision tree on attribution markers and a custom dataset consisting of 43 authentic paintings and 12 forged paintings. It is important to note that the attribution markers consist of typical forensic metrics that are currently used by art historians for art authentication purposes in addition to markers from the painting image [6]. A similar concept to attribution markers involves the forensic technique of optical coherence tomography (OCT), which provides analysis of the cracks in paintings. Both the nature of painting cracks and the map of painting cracks for an authenticated artwork provide quick methods for determining art forgeries [7]. From a legal perspective, an art expert is used to authenticate art using methods of connoisseurship, provenance, and scientific analysis. Art experts are not legally regulated, and the methods are subject to human error. A look into the future indicates that companies, such as Art Recognition, and academic institutions, such as Rutgers University, have proprietary capabilities in detecting intentional forgeries with 80% accuracy, with respect to an undisclosed number of artists; this represents a step forward in eliminating human error with art authentication [8].

Art authentication is paramount for the value of artwork. Traditional art authentication attribute markers are expensive and time-consuming. This research is anticipated to serve as an additional attribute marker, solely from an artist's painting images, to support art authentication, where traditional art authentication methods are inadequate or missing. This attribute marker is used for any artist in the model as a binary attribute. For the prediction of a piece of art for an artist in question, a successful prediction provides a favorable outcome for one artist and unfavorable outcomes for the remaining 2367 artists. Both the accuracy of the prediction and the number of artists being considered are important to indicate that the art in question is properly attributed to the artist in question and not another artist in the model.

## 2. Materials and Methods

Figure 1 demonstrates the process for the creation of an image-only art authentication attribute marker to model 2368 artists. When the process begins, images are partitioned into training, validation, and test sets. To learn the model, the process trains for up to 30 epochs. An epoch is a learning event that includes all paintings in the training set. In each epoch, training paintings are shuffled and mutated to prevent overfitting the model, and the artist's style is gradually learned in batches. At regular intervals, the model is validated using validation paintings, and the results of learning validation make changes to the model, which are used in the next iteration. Once validation results meet a threshold or 30 epochs have passed, the process stops with the current state of the model. This model is used on test paintings to determine the artist. The results of this test produce a confusion matrix, which is a table showing true negatives/positives, false negatives, and false positives. True negatives/positives indicate that the model made a correct negative or positive prediction with respect to the artist and painting in question. False negatives indicate that the model predicted another artist instead of the actual artist. False positives indicate that the model predicted the actual artist, but it should have predicted another artist instead. There are a variety of metrics that can be calculated from this confusion matrix. Equation (3) is used for the primary metric due to the imbalanced nature of painting datasets.

### 2.1. Machine Learning Development and Evaluation

An ML model is developed to predict an artist by training a model using the state-of-the-art ResNet algorithm, to learn relationships between input painting images and corresponding artists that have been labeled manually by visual inspection. Hyperparameters listed in Table 1 are used to continually validate the model being generated until a desired result is achieved or max validation steps occur. The resulting model is evaluated on an unseen test dataset, which is the 20% test partition discussed in the next section.

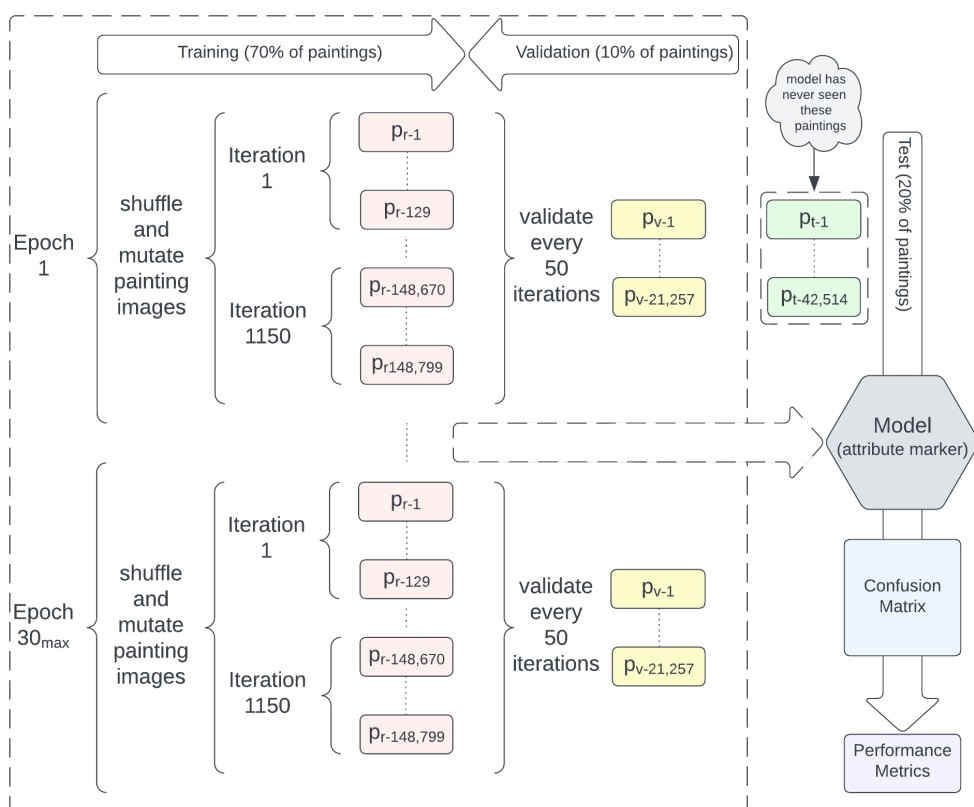

**Figure 1.** Process for image-only art authentication model attribute marker. $p_{ri}$ denotes paintings in the training set. $p_{vi}$ denotes paintings in the validation set. $p_{ti}$ denotes paintings in the test set. Paintings in the training, validation, and test sets are mutually exclusive.

**Table 1.** ResNet Hyper Parameters Used for All Experiment Results.

| Parameter | Value | Purpose |
| --- | --- | --- |
| Image Size | $224 \times 224 \times 3$ | Resize to match network input |
| Training | 70% | Baseline value |
| Validation | 10% | Baseline value |
| Test | 20% | Baseline value; performance measure source |
| Image Rotation | random | prevent overfitting |
| Image Scaling | random | prevent overfitting |
| Image Reflection | random | prevent overfitting |
| Image Batch | 128 | Based on total image count and available resources |
| Maximum Epochs | 30 | Training Governor |
| Validation | 50 iterations | Training Governor |
| Image shuffle | each epoch | Handles indivisible image partition |
| Initial Input Weight | ImageNet TL | Initial weights for neural network |
| Solver | SGDM | Algorithm that updates weights and biases to minimize the loss function |
| Learning Rate | 0.01 | Tuned to ensure training does not take too long or results do not diverge |
| Momentum | 0.9 | Parameter contribution of the previous iteration to the current iteration |
| Weight Decay Regularization | 0.0001 | Reduces overfitting |

*2.2. Training, Validation and Testing Datasets*

An image dataset, sourced from Artfinder, encompasses over 212,570 paintings and is utilized for training, validation, and test sets. Each image adheres to a minimum size of 1200 px × 1200 px and an sRGB color profile is used for training, validation, and test sets. Images are resized using bilinear interpolation into a 224 × 224 × 3 tensor. The training set is randomly generated from 70% of the images and the validation set is randomly generated from 10% of the images. The remaining 20% of images are set aside for testing after the model is trained. An epoch is defined as a single pass through the entire dataset, with images processed in batches of 130–140 images, up to 1050 times. Up to 30 epochs of training and validation occur in a cross-folded fashion every 50 iterations. To mitigate over-fitting concerns, images are shuffled, rotated between −90 and 90 degrees, randomly scaled between 1 and 2 times, and undergo random reflection on the x-axis with each epoch. The process of mutating images in this manner is known as data augmentation and it helps to prevent the network from overfitting and memorizing the exact details of the training images [9]. Transfer learning, given a model from the ImageNet project, is used for additional initialization of training parameters [10]. The ImageNet transfer learning model is used as a starting point for our model. Without this starting point, in past experiments, we found that results took longer to converge and the final accuracy suffered [4].

*2.3. Model Selection*

Existing work from the literature review is leveraged to select a model for training. Features extracted from images using SIFT, HOG, and other digital image processing algorithms consumed by basic ML algorithms, such as support vector machines (SVMs), decision trees, and the k-nearest neighbor algorithm (k-NN) train models quickly, but accuracy starts to suffer quickly as approximately fifty artist classes are approached. Deep neural networks remedy this limitation at the cost of the training time and the need for a high-performance cluster to generate the model. Of these networks, ResNet 101 outperforms earlier versions of ResNet as well as AlexNet, VGG, GoogLeNet, PigeoNet, and CaffeNet [11–17]. Moreover, there may be some performance improvements with SENet and deeper versions of ResNet. However, the scope of this work is not to perform a detailed model comparison or improve upon a model that is working well. The scope of this work is to leverage state-of-the-art models on a very large contemporary art dataset so we can compare these results with existing experiments for fitness and scalability. Therefore, ResNet 101—with an annealing process—is used to produce the models.

*2.4. ResNet Architecture*

The ResNet algorithm solves the exploding and vanishing gradient problem of deep neural networks with a deep residual learning framework, which allows for much deeper networks using the concept of skip connections [18]. A mathematical proof demonstrates how skip connections can largely circumvent the exploding and vanishing gradient problem [19]. ResNet works well with classifying art because the deep network enables multiple passes on an artist's body of work at varying filter sizes, in a generic manner. This process produces a model that does a very good job of learning an artist's style.

*2.5. Artist Selection*

Twenty-four experiments are performed to discover how classification metrics and artist style evolve as the number of artists is reduced. The first experiment is seeded with artists having 10 or more artworks. This provides 2368 artists for the first experiment. The next experiment consists of 2300 artists, and the process continues by reducing the artist count by 100, continuing until the final experiment, which involves 100 artists. The artist selection criteria used to determine which artists will be dropped are based on macro-balanced accuracy. Macro-balanced accuracy is used over micro-balanced accuracy because the metric provides a more granular selection for the fitness of an artist, which results in fewer ties [4].

### 2.6. Evaluation Using the Testing Set

In a final effort to determine the fitness of the model, it is tested on 20% of the paintings split out before training to assure the model has never seen these paintings. Since the number of paintings produced by artists is naturally unbalanced, and a true representation of the artist model without over- and under-sampling is desired, macro-balanced accuracy is calculated from the confusion matrix produced by the test [20,21]. The accuracy ranges from 48.97% for the largest experiment to 91.23% for our smallest experiment. The test accuracy is approximately equal to validation accuracy in all experiments. This indicates the model is not subject to overfitting concerns. Validation accuracy is calculated by Matlab with each validation iteration, which takes into consideration ROC analysis.

### 2.7. Limitations with Image-Based Art Authentication

Several limitations exist with performing art authentication with painting images alone. First and foremost, it is difficult to acquire data. Both physical and online art galleries protect image data because the image is the primary proprietary asset for sale. Access to the complete collection of an art gallery for research purposes requires a trusted relationship with the gallery or a legitimate method of crawling the gallery's online website for image data. Second, there is a varying number of paintings produced by artists, which naturally leads to imbalanced data. The task of gathering more data samples is difficult because the time it takes for an artist to produce new works is nondeterministic. From a sampling perspective, undersampling is not desired because the model does not have the opportunity to learn more about an artist's paintings, and oversampling is not desired because a true representation of the artist's body of work is not obtained. Therefore, metric calculations are used to acquire meaningful multi-class metrics from the tests that assume input classes are not in balance [20,21]. Third, there are attribution markers other than a digital representation of the painting used when authenticating a painting. These markers have traditionally been used by art historians for art authentication. Over 30 attribution markers are discussed in state-of-the-art research dealing with art authentication. For example, there are markers corresponding to the UV, IR, and X-ray physical analysis of a painting. Markers characterizing the pigments and medium characteristics of the artist and time period are considered. Moreover, there are markers that have nothing to do with the actual image, such as signature and ownership documents, as well as history [6]. Fourth, there are no paintings representing true negatives on purpose in experiments.

#### Data Source

The data for experiments come from Artfinder, which is an online art marketplace. Raw data of artwork images were thoroughly reviewed to ensure sound data for experiments. Data from this website were collected over several years via automated web crawling technology [22]. Permission was given to use these data in this research and report aggregate results only. Specific artist names are hidden in this research (hence, the omission of the artist's name and painting images). Upon request, data are available for research verification and extension.

## 3. Results

High-level results for experiments are listed in Table 2. This table represents all 24 experiments, starting with 2368 artists and ending with 100 artists. Validation accuracy (Val Acc) is the accuracy obtained during training. Test accuracy (Test Acc (*M*)) is the primary metric of interest and is the calculated macro-balanced accuracy of the test paintings that were not observed during training. Test accuracy is a bit higher than validation accuracy, which indicates the model did not encounter any overfitting issues during training. The number of paintings observed during the 70/10/20 split is represented by Train/Val/Test Cnt, respectively. The batch size of images used during each iteration of training is represented in the 'Batch' column and the total number of Iterations per epoch is represented by the 'Iterations' header. With each experiment, the average number of

artworks per artist increases proportionally. It is important that this number is increasing to ensure that the model is not influenced by an artist with a more limited number of artworks. Specifically, the average starts at 18 paintings per artist for 2368 artists and ends with 41 paintings per artist for 200 artists. The average dips down to 28 for the last experiment of 100 artists. The first four metrics discussed in this section are listed in Table 2, and the remaining metrics are dropped due to importance and limited space.

**Table 2.** Experiments Results.

| Artists | Val Acc | Test Acc ($\mu$) | Test Acc ($M$) |
|---------|---------|------------------|----------------|
| 2368 | 67.62% | 65.33% | 48.97% |
| 2300 | 68.09% | 66.02% | 50.93% |
| 2200 | 68.67% | 67.20% | 52.88% |
| 2100 | 69.15% | 67.63% | 54.84% |
| 2000 | 69.71% | 68.37% | 57.35% |
| 1900 | 70.49% | 68.95% | 59.35% |
| 1800 | 71.42% | 70.23% | 61.05% |
| 1700 | 72.49% | 71.47% | 63.66% |
| 1600 | 73.29% | 72.76% | 65.34% |
| 1500 | 74.29% | 73.41% | 66.80% |
| 1400 | 75.76% | 74.41% | 68.34% |
| 1300 | 76.66% | 75.93% | 70.51% |
| 1200 | 77.81% | 77.43% | 71.77% |
| 1100 | 78.83% | 78.46% | 74.01% |
| 1000 | 79.59% | 79.57% | 75.40% |
| 900 | 81.34% | 81.57% | 77.20% |
| 800 | 82.49% | 82.35% | 78.36% |
| 700 | 83.75% | 83.35% | 80.33% |
| 600 | 85.59% | 85.71% | 82.66% |
| 500 | 86.46% | 86.85% | 83.60% |
| 400 | 88.15% | 88.51% | 85.47% |
| 300 | 91.11% | 91.30% | 88.88% |
| 200 | 93.17% | 93.36% | 91.15% |
| 100 | 96.20% | 96.29% | 91.23% |

*3.1. Confusion Matrix*

The confusion matrix in Figure 2 represents the largest experiment. Due to the large number of artist classes, a pixel-based confusion matrix is used. The intensity color of each pixel represents the strength of the metric. The diagonal from the upper left-hand corner to the lower right-hand corner in the confusion matrix represents correct predictions in the form of true negative and positive predictions. A distinct, visible diagonal is a favorable condition for the confusion matrix as this will likely indicate a favorable accuracy metric. Horizontal pixels represent false positives and vertical pixels represent false negatives. These pixels are barely visible; this is a favorable condition because it indicates a failed prediction, which minimizes the negative impact on the accuracy metric.

The confusion matrix is also partitioned by the primary art styles of the artists represented. This provides a method to determine which styles are confused. The primary art style for an artist is determined by the largest count of paintings of a given style for the artist. The first alphabetical style is used for ties. For example, "artist 1004" has the following painting styles by count: Impressionistic (13), Expressive and gestural (3), Urban and Pop (3), Abstract (2), Geometric (1), Organic (1), and Photorealistic (1). Therefore, 'Impressionistic' is attributed to "artist 1004". The name "artist 1004" is used because the agreement with Artfinder, the provider of the data, is to keep the artist's name anonymous.

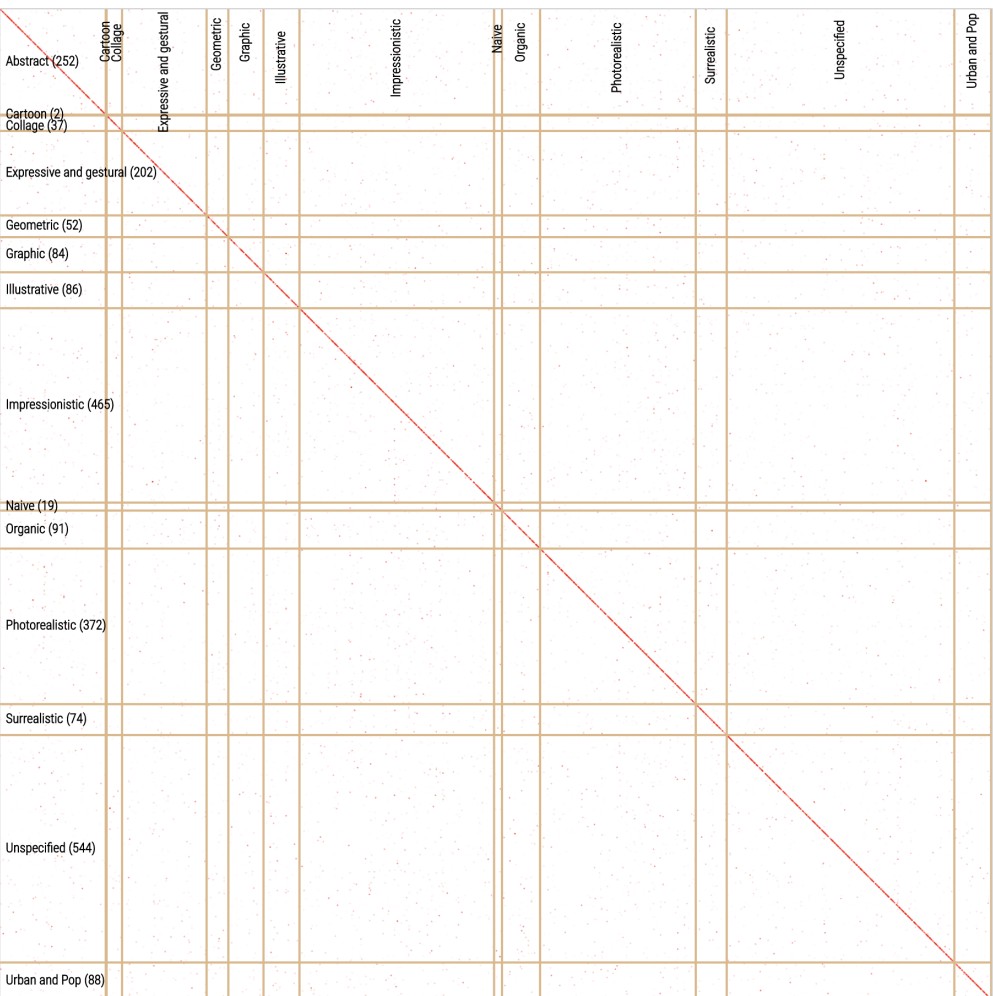

**Figure 2.** Pixel-based confusion matrix of the largest experiment.

### 3.2. Accuracy

The typical average accuracy calculation from the confusion matrix shown in Equation (1) cannot be used because it applies to binary classification [21]. Several techniques are combined from state-of-the-art multi-classification performance measure research to arrive at Equation (3). This equation represents macro-balanced accuracy, which provides reasonable accuracy because it prevents unbalanced majority and minority classes from influencing the overall accuracy [20,21]. The corresponding micro-balanced accuracy shown in Equation (2) is also available. This equation reduces to the average multi-classification recall calculation over all artists; thus, it is not used, even though it provides a better number for reporting. If the data were balanced on the front end of experiments, this metric would be legitimate and would converge with macro-balanced accuracy [20]. Micro-balanced accuracy is reported to demonstrate that it coincides with validation accuracy, which is unbalanced. This demonstrates that the model is not overfitting.

$$\text{AvgAcc} = \frac{\sum_{i=1}^{l} \frac{tp_i + tn_i}{tp_i + tn_i + fn_i + fp_i}}{l} \tag{1}$$

$$\text{BalAcc}_\mu = \frac{\sum_{i=1}^{l} tp_i}{\sum_{i=1}^{l} tp_i + fn_i} = \frac{\sum_{i=1}^{l} tp_i}{\text{Total Predictions}} \tag{2}$$

$$\text{BalAcc}_M = \frac{\sum_{i=1}^{l} \frac{tp_i}{tp_i + fn_i}}{l} \tag{3}$$

## 4. Discussion

### 4.1. Multiclass Classifier as Binary Classifier

In Table 3, the artist count is inversely proportional to validation and test accuracy. Given the state-of-the-art research using multi-classification for image-only art authentication, this behavior is expected [3,4,23]. With these experiments, the goal is to reproduce the counterintuitive phenomenon wherein many classes can improve multi-classification metrics as the number of classes grows [24]. While this phenomenon was not observed, multi-classification for a binary classification art authentication problem is important because training multiple binary classifiers for the artists of interest is not required [25]. Moreover, training a model on more than one artist produces a model of an artist's paintings in addition to what is not considered a painting by the artist in question. The overall model accuracy is reduced in these situations, but a true positive provides more information about the artists to whom the painting in question does not belong. To show this concept in these experiments, consider artist1051, which exists in all experiments. In 16 of the experiments, including the experiment with the most and least artists, the model predicts the artist with 100% accuracy with the test painting data. In three of the experiments, the model predicts the artist with 87.50% accuracy with the test painting data. In five of the experiments, the model predicts the artist with 85.71% accuracy with the test painting data. Given that the accuracy is high in all experiments, the test with the most artists is more meaningful because there are many other potential artists in the model that could confound the test.

**Table 3.** Dataset accuracy comparison.

| Artists | Artfinder Acc | WikiArt Acc | Rijks Acc |
|---|---|---|---|
| 1200 | 71.77% | n/a | 32.40% |
| 1000 | 75.40% | n/a | 40.51% |
| 400 | 85.47% | n/a | 58.60% |
| 300 | 88.88% | n/a | 46.70% |
| 200 | 91.15% | n/a | 81.66% |
| 100 | 91.23% | 72.96% | 72.69% |

### 4.2. True Negatives

Adding purposeful true negatives to experiments would be an interesting addition. This could be accomplished by adding a true negative in the form of a contemporary art forgery. Producing art forgeries is difficult because forgery paintings are difficult to acquire due to the obfuscation of the forgery and the rareness of the forgery event. It is also cost-prohibitive to commission forgeries due to constraints on time and money. True negatives could also be accomplished by keeping a random sample of paintings from the artists that were removed after each experiment's annealing process. While changing the process to include true negatives from the previous experiments is a straightforward task, it is a time-prohibitive task to retrain the models.

### 4.3. Contemporary Art Performance

Performance with the Artfinder contemporary art dataset outperforms previous experiments with historical art datasets from WikiArt and Rijksmuseum. In Table 3, accuracy results are compared using the same macro-balanced accuracy metric. In all cases, the Artfinder experiments outperform WikiArt and Rijksmuseum experiments 10+% [3,4]. This increase in accuracy for contemporary art may be because experiments start with over twice as many artists, and the annealing process can select the best artists for classification once reaching a comparable artist count in previous experiments. It could also be because contemporary art has progressed from historical art in a way that provides more opportunities to learn artistic style.

### 4.4. Artist Style

For all experiments, Figure 3 shows the percentage of artist styles represented. This is important to show because it indicates a variety of art styles are represented (from the first experiment with 2368 artists to the last experiment with 100 artists). The evolution of painting style representations demonstrates that models do not favor a specific art style.

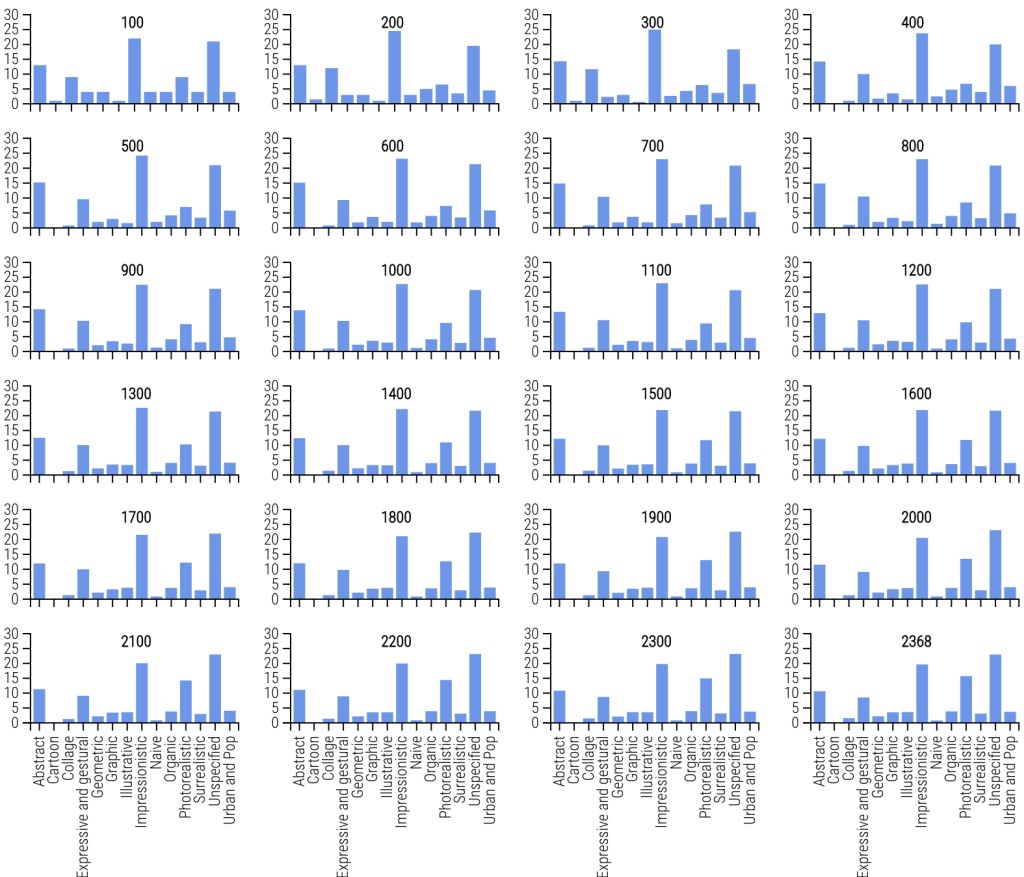

**Figure 3.** Artist Style Evolution.

### 4.5. Uniqueness

This research aims to maximize the number of artists and the accuracy in these experiments. Maximizing both numbers yields the best accuracy with respect to as many artists as possible. It is helpful to have one metric that considers both numbers. Therefore, a uniqueness score is used as a metric to analyze experiments in terms of a ratio between the accuracy, artist count, and the number of paintings. The number of artists per painting, referred to as artist density, is taken into consideration. The artist's painting density is chosen as the numerator and accuracy as the denominator because the focus is on how the artist painting density is distributed in relation to accuracy. In this manner, artist painting density approaches accuracy, in which case, the relative paintings uniquely define the artist. Uniqueness is calculated as the ratio of the artist painting density to accuracy, as seen in Equation (4). As hoped, the uniqueness score is not always proportional to the artist count in the experiments, as seen in Table 4; the ideal number of artists with respect to accuracy is 300. Therefore, an ideal artist count of 300 is recommended for model generation.

$$\text{Uniqueness} = \frac{\text{Artist Count}/\text{Painting Count}}{\text{Accuracy}} \qquad (4)$$

**Table 4.** Uniqueness.

| Artists Count | Uniqueness Score |
| --- | --- |
| 2368 | 11.37% |
| 2300 | 10.76% |
| 2200 | 10.04% |
| 2100 | 9.32% |
| 2000 | 8.57% |
| 1900 | 8.00% |
| 1800 | 7.53% |
| 1700 | 6.99% |
| 1600 | 6.58% |
| 1500 | 6.19% |
| 1400 | 5.82% |
| 1300 | 5.56% |
| 1200 | 5.29% |
| 1100 | 4.85% |
| 1000 | 4.55% |
| 900 | 4.24% |
| 800 | 3.99% |
| 700 | 3.75% |
| 600 | 3.41% |
| 500 | 3.12% |
| 400 | 2.95% |
| 300 | 2.59% |
| 200 | 2.68% |
| 100 | 3.99% |

## 5. Conclusions

In this paper, we contribute toward art authentication research using contemporary art from the Artfinder dataset by applying a performance annealing residual neural network to produce 24 baselines for the dataset. We demonstrate that it is possible to use ML-generated models to authenticate contemporary art from 2368 to 100 artists with an accuracy of 48.97% to 91.23%, respectively. We also provide a method to address very large confusion matrices that consider style and a uniqueness calculation, which provides a method to choose an optimal artist count for model generation.

*Future Work*

In future work, we would like to continue to explore new art datasets as they become available. We would like to compare these results with the results of the dataset in this paper and collaborate with artists to provide more detailed analyses for the art community. We also believe that exploring adversarial attacks on art authentication and investigating saliency maps in art authentication will be of interest.

**Author Contributions:** Conceptualization: T.D. and Z.R.; Data curation: T.D. and Z.R.; Formal analysis: T.D.; Funding acquisition: T.D. and Z.R.; Investigation: T.D.; Methodology: T.D.; Project administration: T.D.; Resources: Z.R.; Software: T.D.; Supervision: T.D. and Z.R.; Validation: T.D.; Visualization: I.C., T.D. and A.-A.-R.N.; writing—original draft preparation: T.D.; Writing—review & editing: I.C., T.D., A.-A.-R.N. and Z.R. All authors have read and agreed to the published version of the manuscript.

**Funding:** This research was partially supported by the National Science Foundation under grant IIP 1749105.

**Data Availability Statement:** The data and code for the training, validation, and test classes are available on GitHub via https://github.com/btdobbs/Contemporary-Art-Authentication-with-Large-Scale-Classification (accessed on 6 October 2023).

**Acknowledgments:** We would like to acknowledge the Technology Solutions Office (TSO) group at the University of North Carolina's Charlotte College of Computing and Informatics for their assistance with this research.

**Conflicts of Interest:** The authors declare no conflict of interest.

## Abbreviations

The following abbreviations are used in this manuscript:

| | |
|---|---|
| ML | machine learning |
| SVM | support vector machine |
| k-NN | k-nearest neighbor |
| PCA | principal component analysis |
| OCT | optical coherence tomography |

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
