# Peer review of "Contemporary Art Authentication with Large-Scale Classification"

_2504-2289, doi:10.3390/bdcc7040162_

Round 1

Reviewer 1 Report

Overall

The authors claim to be able to capture the visual uniqueness of style using a ML mathematical model which can be used for art authentication and report accuracy rates from around 50% to 90% in their experiments. The method described seems well thought through justification of choices is not always provided. I have a number of major and minor issues for the authors to address. However, none of these involve additional experimentation. All of them relate to crafting a more readable paper. The authors should be able to address these rather quickly.

Major issues

1. Situating the research

The introduction is clear and concise, but does not arm readers with the background knowledge necessary to follow this specific research. Readers with no prior knowledge of art authentication do not process a schema to understand how this research fits in with existing methods, and so the methods of identifying should be explained in slightly more detail.

2. Art authentication

Authentication can be conceived as comprising profiling (e.g. identifying the genre, period, etc.), attribution (e.g. identifying the specific artist) or verification (e.g. deciding whether a specific work can was created by a known artist). How is your work situated in related to these types of authentication?

3. Literature review/related works

This is a niche area and so the specific literature will not be as extensive as areas such as image processing. However, art authentication can be considered as a niche subset with similarity detection, and analogical to authorship analysis. A few works are mentioned in the introduction, but there is scant detail, leaving readers to wonder how this work fits in with the extant research.

4. Figure 1 and its description

The figure is clear, but this should be in the methods rather than the introduction.  Many introductions include details on the paper’s contribution to the literature, but to me the text between Lines 40 to 64 reads more like the method.

5. Evidence supporting decisions in method. Cite or explain. e.g.

5a the use of rotation and reflection to mitigate over-fitting

5b ImageNet transfer learning model is used as a optimal starting point

6. Scope

You state “However, the scope of this work is not to perform a detailed model comparison or improve upon a model that is working well.” What do you mean by this statement – you want to improve a model that does not work well rather than improve one that works well? Most readers will assume you want to create a model that works better than others.   Since your introduction is rather sparse, the scope is already unclear. Reading between the lines, I am guessing that you want to produce a model with few false positives that can cope with a large number of potential artists.

7. Figure 2

The contrast between the red dots and white background is insufficient. Even enlarging to 150% I can barely distinguish the dots – I assume the eyesight of the authors is better than mine, but I suspect other readers will have the same issue. Choose colours for the background and details that provide sufficient contrast.

8. Conclusions

This paragraph seems like an afterthought. Here you should state the core contribution of the paper, which should be derived from the content of the discussion immediately prior.

Minor/Language issues

1. Insert citations before the period. (See Lines 21 and 24, and many more)

2. hyper parameter – hyperparameter (Line 70)

3. equation 4 – use a diagonal bar between Artist count and Painting count

Reviewer 2 Report

This study brings a rather modest contribution to the scientific field, investigating primarily applications. But the work done is done correctly, achieving the intended results.

1. Introduction, page 2, penultimate paragraph - unnecessary descriptions of well-known notions such as: learning era, matrix and its elements. These terms do not need additional description as they are widely known in the scientific community.

2. Fig 1 is a strange scheme. It is not completely obvious what exactly the authors wanted to describe with this image. There is a lot of information describing various aspects of the learning process. This includes the number of epochs, the proportion of training/testing split, notes on what validation is, and a mention of the general course of the experiment. All this information does not look structured, does not bring additional insight, and clearly needs to be improved or eliminated.

3.         Section 2.1, when mentioning "recommended hyperparameters", it is desirable to mention which hyperparameters were used (see if they were proposed in the original paper).

4.         Section 2.4. The ResNet architecture is very popular (its mentioning is still found in any review articles, e.g. https://arxiv.org/pdf/2004.02806.pdf). Additional description of its features seems unnecessary, since these unique solutions are not important for the main object of study of this article

5.         Section 2.5. at the beginning of the section mentions 24 experiments using different numbers of authors. But it goes on to describe the number of authors for only two experiments. It is not obvious how the remaining 22 experiments were conducted.

6.         Section 3. It is worth graphing the dependence of the value of the metric on the size of the dataset. This would allow more judgment about the nature of these values.

7.         Fig 2. I certainly realize that it is difficult to display information about such a large number of classes on a diagram of any kind, but the proposed version clearly fails to do so. In order to see the mentioned pixels in the confusion matrix I had to zoom in twice on the laptop screen, focusing on a small part of the image. But this did not provide any useful information in the context of the article.

8.         Please, in a separate section, describe what is unique about this work. How the results obtained are better than others, and what novelty and practical significance they have.

Round 2

Reviewer 1 Report

Overall

The authors have addressed all the major issues raised in my initial review. I no longer have any objections to publication and look forward to seeing this paper in press. By the way, to create a diagonal fraction line in the numerator, consider using the usepackage called x diagfrac.

Reviewer 2 Report

I have read the second reviewer's comments and I don't see where they conflict. But in general, the work meets the requirements and can be published.